# *Funneliformis mosseae* Improves Growth and Nutrient Accumulation in Wheat by Facilitating Soil Nutrient Uptake under Elevated CO_2_ at Daytime, Not Nighttime

**DOI:** 10.3390/jof7060458

**Published:** 2021-06-07

**Authors:** Songmei Shi, Xie Luo, Miao Wen, Xingshui Dong, Sharifullah Sharifi, Deti Xie, Xinhua He

**Affiliations:** 1Centre of Excellence for Soil Biology, College of Resources and Environment, and Chongqing Key Laboratory of Plant Resource Conservation and Germplasm Innovation, School of Life Sciences, Southwest University, Chongqing 400716, China; shismei@email.swu.edu.cn (S.S.); luox123@email.swu.edu.cn (X.L.); wenmiao77@163.com (M.W.); xingshuid@outlook.com (X.D.); nsharifullah@gmail.com (S.S.); xdt@swu.edu.cn (D.X.); 2National Base of International S&T Collaboration on Water Environmental Monitoring and Simulation in Three Gorges Reservoir Region, Chongqing 400716, China; 3Department of Land, Air and Water Resources, University of California at Davis, Davis, CA 95616, USA; 4School of Biological Sciences, University of Western Australia, Perth, WA 6009, Australia

**Keywords:** arbuscular mycorrhiza, biomass production, enzyme activity, nitrogen, phosphorus, potassium

## Abstract

The concurrent effect of elevated CO_2_ (eCO_2_) concentrations and arbuscular mycorrhizal fungi (AMF) on plant growth, carbon (C), nitrogen (N), phosphorus (P) and potassium (K) accumulations in plant and soil is largely unknown. To understand the mechanisms of eCO_2_ and mycorrhization on wheat (*Triticum aestivum*) performance and soil fertility, wheat seedlings were grown under four different CO_2_ environments for 12 weeks, including (1) ambient CO_2_ (ACO_2_, 410/460 ppm, daytime/nighttime), (2) sole daytime eCO_2_ (DeCO_2_, 550/460 ppm), (3) sole nighttime eCO_2_ (NeCO_2_, 410/610 ppm), and (4) dual or continuous daytime/nighttime eCO_2_ ((D + N)eCO_2_, 550/610 ppm), and with or without AMF (*Funneliformis mosseae)* colonization. DeCO_2_, NeCO_2_ and (D + N)eCO_2_ generally significantly increased shoot and root biomass, plant C, N, P and K accumulation, soil invertase and urease activity, but decreased shoot and root N, P and K concentrations, and soil available N, P and K. Compared with non-AMF, AMF effects on above-mentioned characteristics were significantly positive under ACO_2_, DeCO_2_ and (D + N)eCO_2_, but negative on plant biomass, C, N, P and K accumulation under NeCO_2_. Overall, AMF colonization alleviated soil nutrient constraints on plant responses to DeCO_2_, while NeCO_2_ decreased AMF’s beneficial effects on plants. These results demonstrated that an integration of AMF’s benefits to plants under factual field DeCO_2_ and/or NeCO_2_ will be critical for managing the long-term consequence of future CO_2_ rising on global cropping systems.

## 1. Introduction

An increase in atmospheric carbon dioxide (ACO_2_) concentration is one of the most important environmental factors reflecting global climate change [1]. The ACO_2_ concentration has been increased from 280 ppm during the industrial revolution to 419.64 ppm (https://www.co2.earth Accessed on 5 June 2021), and could reach ~550 ppm in the next 50 years [1]. Elevated CO_2_ (eCO_2_) directly influences soil–plant systems via improving plant growth [2,3,4]. Elevated CO_2_ generally exhibits ‘fertilization effects’ due to stimulation of photosynthesis and biomass accumulation in various C_3_ crops, including wheat (*Triticum aestivum* L.) [5,6,7]. Increased productivity of crops needs a large supply of nutrients (especially nitrogen, N; phosphorous, P; and potassium, K; etc.) to match their increased carbon (C) assimilation under eCO_2_ [8,9,10,11]. Soil nutrient availability was indeed decreased over a long-term eCO_2_ owing to an increased nutrient demand by eCO_2_-stimulated growth [12]. The eCO_2_ effects were often gradually diminished when plants grew under nutrient-limited soils [13,14]. Furthermore, the beneficial effect of eCO_2_ on biomass production can lead to changes in C, N, P, K, and other nutrient content in both plants and soils [15,16,17,18]. These changes affected agricultural ecosystem processes, including nutrient cycling [8], soil organic matter decomposition [19], and microbial processes [20,21]. Soil enzymes, which integrate information from soil microbial and biochemical statuses, could serve as indicators of various changes in the plant–soil system [22]. Thus, soil enzyme activities are also likely to change under eCO_2_ [3,23]. It would be worthy to understand the response of aboveground biomass and the feedback of belowground components (especially soil nutrients, soil microorganisms and enzyme activities) to eCO_2_.

Arbuscular mycorrhizal fungi (AMF) can form mutualistic associations with the roots of >80% of vascular plant species and improve plant nutrient uptake [24]. AMF are important in defining plant responses to eCO_2_ [4,25,26,27,28]. Elevated CO_2_ at daytime or continuously from daytime to nighttime has a positive effect on C_3_ crops’ photosynthetic C assimilation, which increases the belowground transportation of photosynthates to support AMF symbiosis [29,30]. In turn, AMF help to alleviate the increased plant nutrient limitation associated with increased photosynthetic rates under daytime or continuous eCO_2_ [27,30,31]. The costs and benefits to plants and AMF are a function of the balance between the C cost of fungi and nutrient supply to the plant. On the one hand, eCO_2_ usually decreases N, P, and K concentrations in plant tissues [15,17,32], such as a decrease of 27% for N, 34% for P, and 20% for K in wheat grains [33], and 29% for N in wheat flag leaves [34]. On the other hand, eCO_2_ improves plant N, P, and K acquisition when plants associate with AMF [4,26,35,36], which help to alleviate N-, P-, or K-limitation [10]. However, the effects of mycorrhizal association are not always positive under eCO_2_ when the C costs of AMF outweigh their benefits to plant nutrient uptake [37]. For instance, N availability to flag leaves of the *Rhizophagus intraradices* colonized durum wheat was lower under 700 ppm eCO_2_ than under 400 ppm CO_2_ [34]. The mechanisms for AMF on plant performance under daytime eCO_2_ thus remain largely unknown, let alone under nighttime eCO_2_.

In fact, the atmospheric CO_2_ concentration at plant height is usually higher during nighttime than during daytime [38], considering the corresponding CO_2_ variation or fluctuation with plant photosynthesis and soil respiration, particularly in agricultural fields. The average atmospheric CO_2_ in Australia, Japan and the USA varied from 390 ppm during daytime to 465 ppm during nighttime [38]. Indeed, a three-year (2017–2019) period of our field observation recorded that the average daily atmospheric CO_2_ ranged from 417 ± 16 ppm at daytime and 463 ± 27 ppm at nighttime at the National Monitoring Base for Purple Soil Fertility and Fertilizer Efficiency close to the campus of Southwest University, Chongqing, China (see Appendix A from October 2017 to March 2018). As a consequence, plants should differentially respond to such contrasting daytime or nighttime atmosphere CO_2_ concentrations [39]. Thus variations in daytime and/or nighttime atmosphere CO_2_ concentrations shall provide a closer simulation of currently atmospheric CO_2_ conditions that plants will respond to in the near future. However, owing to the likely extra cost and maintenance of CO_2_ gas supply, only a few studies have examined the different responses of plant growth and yield between contrasting daytime and nighttime eCO_2_ concentrations [40,41,42,43,44]. For instance, it was daytime, not nighttime eCO_2_, that improved *Morus alba* growth [39]. The biomass production in *Amaranthus retroflexus* and *Zea mays* was significantly reduced under 700 ppm nighttime eCO_2_ than under 370 ppm nighttime eCO_2_ [45], although such nighttime eCO_2_ effects on plant growth were plant species-specific [46]. The contrasting positive or negative effects of nighttime eCO_2_ on plant growth would result from dark respiration [47,48]. It is suggested that dark respiration was decreased by 20–45% under 700–1000 ppm eCO_2_ at nighttime in the short-term [41,43,49]. In contrast, dark respiration usually, but not always, increased in the long-term such CO_2_ nighttime enrichments [47,50]. Mechanisms of eCO_2_ at nighttime that may affect plant growth and C assimilation have not been established yet. Questions hence arise as to whether the plant nutrient demand could be increased under eCO_2_ at nighttime, and whether AMF symbiosis could play the same role in C and nutrient balance under eCO_2_ at daytime or nighttime. Such information is essential for understanding the mechanisms affecting C and N dynamics under future CO_2_-increasing scenarios.

We therefore designed an environment-controlled system, which has minimal impact on light, air temperature and humidity, while providing either on-site ambient or elevated CO_2_ concentrations for growing plants during daytime and/or nighttime. The objectives of the present study were to address: (1) How auto-controlled field daytime and/or nighttime eCO_2_ could affect plant biomass production, and C, N, and P uptake or accumulation; (2) whether AMF effects on soil nutrient uptake could alleviate nutrient constraints on responses to eCO_2_ at both daytime and nighttime; and (3) whether the interactive effects of eCO_2_ and AMF on plant performance and related soil properties could be differentiated between daytime and nighttime eCO_2_. In doing so, winter wheat (*T**. aestivum* cv. Yunmai) inoculated with or without AMF was grown in soil (Eutric Regosol, FAO Soil classification system) filled pots inside environmentally controlled glass-made chambers, which had similar growth conditions except different CO_2_ concentrations at daytime and/or nighttime. Plant performance and soil properties were then compared 12 weeks after sowing.

## 2. Materials and Methods

### 2.1. Experiment Design and Treatments

In a completed random arrangement, the experiment was a split plot design with atmospheric CO_2_ concentrations as the main factor and mycorrhizal inoculation as the subfactor, and involved two AMF treatments: inoculated with *Funneliformis mosseae* and autoclaved *F. mosseae* (non-AMF) and four atmospheric CO_2_ concentration treatments (Figure 1). Based on the on-site daily observations of 417 ± 16/463 ± 27 ppm (daytime/nighttime) between October 2017 and March 2018 (see Appendix A) and an estimated ~550 ppm in the next 50 years [1], four different day and/or night CO_2_ concentrations (±30 ppm) or treatments were applied: (1) ambient CO_2_ (ACO_2_, 410 ppm daytime/460 ppm nighttime), (2) daytime eCO_2_ only (DeCO_2_, 550/460 ppm), (3) nighttime eCO_2_ only (NeCO_2_, 410/610 ppm), and (4) continuous daytime and nighttime eCO_2_ [(D + N)eCO_2_, 550/610 ppm]. The respective daytime and nighttime eCO_2_ concentrations were thus increased by ~33.33% of the ACO_2_ treatment. Daytime was from 07:00 a.m. to 19:00 p.m. and nighttime was from 19:00 p.m. to 07:00 a.m. Each CO_2_ treatment owned three chambers for a total of 12 chambers to the four CO_2_ treatments in a completely randomized experimental arrangement within three blocks (Appendix A). The arrangement of chambers was crisscrossed with 4 m apart from each other to avoid the sunshade of chambers.

### 2.2. Experimental Facility

This study was conducted from 9 November 2017 and 2 February 2018 in a CO_2_ exposure facility at the National Monitoring Base for Purple Soil Fertility and Fertilizer Efficiency (29 °48 ′ N, 106 °24 ′ E, 266.3 m above sea level) on the campus of Southwest University, Chongqing, China. The CO_2_ auto-controlling facility (DSS-QZD, Qingdao Shengsen Institute of CNC Technology, Shandong, China) consists of a control system and 12 environmentally controlled chambers (Appendix A), and the CO_2_ is supplied by CO_2_ cylinders (Appendix A). The CO_2_ cylinders with electric point pressure meters are connected to a CO_2_ control system to maintain CO_2_ gas flow into each chamber with the targeted CO_2_ concentration (Appendix A).

Each growth chamber had a rectangular floor base, supported by a steel frame hanging 50 cm above the cement ground base (Appendix A). The bottom floors of the growth chamber are made up of polyvinyl chloride plates, and the four-sided walls and top roofs of the chamber are constructed by tempered glass (10 mm thickness, 90% light transmission rate, Yutao Glass Company, Jiulongpo, Chongqing, China) (Appendix A). The growth chamber has a size of 1.5 m × 1.0 m × 2.5 m (length × width × height) in order to grow maize, *Sorghum bicolor*, *Glycine max*, wheat, and so forth. The electron sensors for monitoring humidity, temperature, light intensity, and CO_2_ concentrations are mounted on the outer and inner surfaces of the glass wall in each chamber to monitor their variations (Appendix A–D). The air humidity, temperature, and CO_2_ concentrations are automatically controlled by their respective electronic bits and pieces (Appendix A). The monitor’s signals are fed into proportionally integrated differential controllers that regulate the opening time within a 10 s cycle (Appendix A). This automatic electronic controlling system can automatically regulate and instantly visualize the fluctuation of ±30 ppm CO_2_ concentration, ±0.5 °C air temperature and ±5% humidity inside and outside the chamber (Figure 2 and Appendix A). The targeted CO_2_ concentration inside is maintained by injecting 99.99% CO_2_ from the cylinder (Appendix A) using a solenoid valve controlled by a mini-computer (Appendix A). When the CO_2_ concentration inside a chamber exceeds the targeted concentration, the inside air is pumped out using a pump controlled by the mini-computer and filtered with 1.0 M NaOH solution. When the humidity inside a chamber is higher than that of outside the air humidity, the inside air is pumped out using another pump controlled by the mini-computer and filtered with solid anhydrous calcium chloride. The temperature is automatically maintained at 0.5 °C variation between inside and outside the chamber using an air conditioner (Gree, Zhuhai Gree Corp., Zhuhai, China) controlled by the mini-computer.

### 2.3. Mycorrhizal Inoculum, Growth Soil and Plant Growth Conditions

The inoculum of AMF (*Funneliformis mosseae*) was purchased from the Bank of Glomales at the Institute of Plant Nutrition and Resources, Beijing Academy of Agriculture and Forestry, Beijing, China. The inoculum was a mixture of soil (50 spores per gram dry soil), mycorrhizal mycelia and root segments. The growth soil (Eutric Regosol, FAO Soil Classification System, developed from Jurassic purple shale and sandstone) was air-dried, sieved by passing through a 2 mm mesh and sterilized at 121 °C for 120 min. The pots (height/diameter = 21/17 cm) were then filled with 3.4 kg of sterilized soil. The soil (pH 6.8) had 10.56 g of organic carbon kg^−1^, 0.66 g total N kg^−1^, 0.61 g total P kg^−1^, 97 mg available N kg^−1^, 17 mg available P kg^−1^ and 197 mg available K kg^−1^.

Seeds of winter wheat (*T**. aestivum* cv. Yunmai) were surface-sterilized with 10% H_2_O_2_ for 20 min, thoroughly rinsed with sterile water, and then pre-germinated on sterilized moist filter paper at 25/20 °C (day/night) for 36 h. Eight germinating wheat seeds were sown in one plastic pot. A total of 20 g of *F. mosseae* inocula were put at a 5 cm soil surface depth inside each pot, while an equal amount of autoclaved (121 °C, 0.1 Mpa, 120 min) inoculum was supplied to the non-mycorrhizal pots. A volume of 5.0 mL filtrate (0.45 µm syringe filter, Millipore Corporation, Billerica, MA, USA) from the *F. mosseae* inoculum was added to each non-mycorrhizal pot to minimize differences in other microbial communities. Then, two mycorrhizal pots and two non-mycorrhizal pots were placed into each growth chamber, and thus, the three replicated chambers had a total of six replicated pots for each CO_2_ concentration treatment. Except the CO_2_ concentration, the chambers had similar other growth conditions, such as light, air temperature and humidity, as monitored by the above-mentioned auto-controlling facility (Figure 2). To minimize differences in growth conditions, the positions of growth pots in each chamber were rotated once a week, and shifted to another replicate chamber once fortnightly. In addition, all the pots with plants were watered once with Hoagland solution to a total of 100 mg N, 50 mg P and 75 mg K per pot and the soil moisture during the whole growth period was maintained at 70% water-holding capacity with sterilized water by routing weighing of pots once every two days.

### 2.4. Harvest, Sampling and Analyses

Plant and soil samples were harvested 12 weeks after sowing during the jointing stage and were combined from the two pots in each chamber as a composite sample. Plant tissues were divided into shoots (leaves and stems) and roots. Plant fresh roots were carefully washed with tap water and rinsed with deionized water. A portion of fresh roots was stored in 50% ethanol to determine root AMF colonization. The remaining fresh roots and shoots were dried at 105 °C for 30 min and then at 75 °C for >48 h until they reached a consistent dry weight. Soil samples, collected from soils that had been well-mixed from each growth pot, were divided into two parts after the removal of debris and fine roots. The first part of the soil was air-dried for >48 h for the determination of chemical properties, and the second part was immediately transferred to the laboratory and stored at −20 °C for the determination of enzyme activities.

### 2.5. Determination of AMF Colonization

The percentage of root AMF colonization was measured according to Brundrett et al. [51]. The roots were cut into 1.0 cm segments and cleared with 10% (*w*/*v*) KOH in a water bath at 90 °C for 20 min, rinsed in water. The cleared root segments were acidified in 0.2 M HCl for 3 min and then stained with 0.05% trypan blue. The stained segments were mounted on glass slides, and a total of 50 randomly selected root segments from each replicate were examined under a microscope. The ratio of the number of root segments that showed a fungal structure (spores, hyphae, arbuscules or vesicles) and the total number of root segments was calculated as the percentage of root AMF colonization.

### 2.6. Determination of C, N, P and K in Plants and Soils

The oven-dried shoot and root and air-dried soil samples were ground to fine powder for soil and plant C, N, P and K analyses. Plant C concentration was determined using the potassium dichromate–sulfuric acid oxidation method [52]. After digestion with 98% sulfuric acid and 30% hydrogen peroxide, plant N, P, and K concentrations were determined with the Kjeldahl method, the vanadium molybdate yellow colorimetric method, and flame photometry, respectively [52]. Soil available N (AN) was measured by the micro-diffusion technique after alkaline hydrolysis [52]. Soil available P (AP) was extracted with 0.5 M NaHCO_3_ and then measured by the Mo-Sb anti spectrophotometric method [52]. Soil available K (AK) was extracted with 1.0 M ammonium acetate and then determined by flame photometry [52].

### 2.7. Determination of Soil Enzyme Activity

Soil invertase activity (mg glucose g^−^^1^ soil h^−^^1^) was determined firstly by incubating five grams of fresh soil with 1 mL toluene, 15 mL 8% (*w*/*v*) sucrose, and 5 mL phosphate buffer (pH 5.5) for 24 h at 37 °C. After incubation, 1 mL filtrate with 3 mL 3,5-dinitrosalicylic acid were then incubated in boiling water for 5 min. Subsequently, the reaction solution was diluted to 50 mL with distilled water and spectrophotometrically measured at 508 nm [53].

Soil urease activity (mg NH_4_^+^–N g^−^^1^ soil h^−^^1^) was determined firstly by incubating five grams of fresh soil with 1 mL toluene, 10 mL 10% urea solution (*w*/*v*), and 20 mL citrate buffer (pH 6.7) for 24 h at 37 °C. After incubation, 2 mL filtrates were mixed with 4 mL sodium phenol solution and 3 mL 0.9% (*v*/*v*) sodium hypochlorite solution in a 50 mL volumetric flask. After 20 min, the reaction solution was then diluted to 50 mL with distilled water and spectrophotometrically measured at 578 nm [53].

Neutral phosphatase activity (mg phenol g^−1^ h^−1^) was determined firstly by incubating five grams of fresh soil, 1 mL toluene and 5 mL disodium phenyl phosphate solution and 5 mL citrate buffer (pH 7.0) for 24 h at 37 °C. After incubation, 1 mL filtrate with 5 mL borate buffer (pH 9.0), 3 mL 2.5% potassium ferrocyanide (*w*/*v*) and 3 mL 0.5% 4-aminoantipyrine (*w*/*v*) were then thoroughly mixed in a 50 mL volumetric flask. Subsequently, the reaction solution was diluted to 50 mL with distilled water and spectrophotometrically measured at 570 nm [54].

### 2.8. Statistical Analysis

Statistical analysis was performed using SPSS 19.0 software (SPSS Inc., Chicago, IL, USA). Data were shown as mean ± standard error (SE). All response variable data (except for root colonization) were analyzed by two-factor analyses of variance (ANOVA). The factors in the two-way ANOVA were CO_2_ level and arbuscular mycorrhiza. Significant differences among treatments were compared by Tukey’s Multiple Range Test at *p* < 0.05 using SPSS 19.0 software. Graphs were plotted using OriginPro2018 software (OriginLab Corp., Northampton, MA, USA).

## 3. Results

### 3.1. Mycorrhizal Colonization

A significantly greater percentage of root AMF colonization among CO_2_ treatments (*p* < 0.05) ranked in AMF plants as (D + N)eCO_2_ (53.60 ± 2.25) > DeCO_2_ (48.31 ± 3.04) ≈ NeCO_2_ (47.07 ± 3.42) > ACO_2_ (40.67 ± 2.59), whereas no AMF colonization was detected in non-AMF plants.

### 3.2. Effects of AMF and CO_2_ on Plant C, N, P and K Concentration

C concentrations in shoots were significantly higher under eCO_2_ than under ACO_2_ in non-AMF 12-week-old wheat seedlings, but no significant differences among DeCO_2_, NeCO_2_, and (D + N)eCO_2_. In contrast, significantly higher shoot C concentrations ranked in AMF plants as (D + N)eCO_2_ > DeCO_2_ ≈ ACO_2_ > NeCO_2_. Meanwhile, only (D + N)eCO_2_ increased root C concentrations in both non-AMF and AMF plants. However, neither shoot C nor root C concentrations were affected by the *F. mosseae* inoculation and by CO_2_ × AMF interaction (Table 1).

Elevated CO_2_ generally significantly decreased N in both the shoots and roots of AMF and non-AMF plants (Table 1), and such decreases were more pronounced under NeCO_2_ in AMF plants, and DeCO_2_ in non-AMF plants (Table 1). Reduction of shoot P concentrations were observed under both NeCO_2_ and (D + N)eCO_2_ in non-AMF plants and under NeCO_2_ in AMF plants only. Both DeCO_2_ and NeCO_2_ decreased root P concentrations in both AMF and non-AMF plants. K concentrations in the shoot and root were generally significantly lower under eCO_2_ than under ACO_2_ in both AMF and non-AMF plants (Table 1). Meanwhile, *F. mosseae* inoculation significantly increased root N concentrations under both ACO_2_ and DeCO_2_, leaf N concentrations under DeCO_2_ and root P concentrations under DeCO_2_ and (D + N)eCO_2_, and shoot K concentration under NeCO_2_. A significant CO_2_ × AMF interaction was observed in N concentrations in both shoots and roots, and the K concentration in roots, but not for P concentrations (Table 1).

### 3.3. Effects of AMF and CO_2_ on Plant Biomass Production

Elevated CO_2_ increased shoot, root and total plant biomass, regardless of whether the 12-week-old wheat seedlings were inoculated with *F. mosseae* or not (Figure 3A–C). Compared to the non-AMF plants, AMF colonization increased shoot and total plant biomass production, under ACO_2_, DeCO_2_ and (D + N)eCO_2_, but not under NeCO_2_ or for root biomass production (Figure 3B). Meanwhile, a significant CO_2_ × AMF interaction was found for both the shoot and total plant biomass production (Figure 3A,C), but not for root biomass production (Figure 3B).

### 3.4. Effects of AMF and CO_2_ on Plant C, N and P Accumulations

In general, eCO_2_ significantly increased C accumulations in the shoot, root and total plant of the 12-week-old wheat seedlings (Figure 3D–F). The C accumulations in the shoot and total plant were significantly affected by *F. mosseae* colonization and CO_2_ × AMF interaction (Figure 3D,F), but not in root C accumulations (Figure 3E). The accumulations of C in the shoot and total plant were significantly higher for the AMF plants under ACO_2_, DeCO_2_ and (D + N)eCO_2_, but lower for the AMF plants under NeCO_2_, compared to the respective non-AMF plants (Figure 3D–F).

Generally, eCO_2_ significantly increased N accumulation in the shoot and total plant (Figure 3G,I), but not in the root (Figure 3H). Compared with non-AMF wheat plants, AMF plants had greater N accumulation in the shoot, root, and total plant under ACO_2_, DeCO_2_, and (D + N)eCO_2_ (Figure 3G–I), but lower under NeCO_2_ (Figure 3G,H). A significant CO_2_ × AMF interaction on N accumulation was thus observed in the shoot, root, and total plant (Figure 3G–I).

Compared with ACO_2_, eCO_2_ significantly enhanced P accumulation in the shoot by 35–95%, and in the total plant by 30–79% in both non-AMF and AMF plants (Figure 3J,L), but had no effects on root P accumulation (Figure 3K). Neither the AMF symbiosis nor the CO_2_ × AMF interaction showed a significant effect on the P accumulation in the shoot, root, and total plant (Figure 3J–L).

K accumulations in the shoot and total plant were significantly higher under all eCO_2_ treatments in non-AMF plants, and under DeCO_2_ and (D + N)eCO_2_ in AMF plants, compared with ACO_2_ (*p* < 0.05, Figure 3M,O). Significantly higher K accumulations in the shoot and total plant were in AMF than in non-AMF plants under ACO_2_, DeCO_2_, and (D + N)eCO_2_. A significant CO_2_ × AMF interaction on K accumulation was thus observed in the shoot and total plant (Figure 3M,O), but not in root K accumulations (Figure 3N).

### 3.5. Effects of AMF and CO_2_ on Soil Nutrients

Soil AN, AP, and AK were significantly affected by CO_2_, AMF, and CO_2_ × AMF interaction, regardless of whether plants were colonized with AMF or not (Figure 4A–C). Compared with ACO_2_, soil AN, AP, and AK were significantly decreased under all eCO_2_ treatments in non-AMF soils, except AP under NeCO_2_, and under DeCO_2_ in AMF soils. Moreover, soil AN, AP, and AK were significantly increased by 5–36%, 7–49%, and 3–31% in AMF than in non-AMF soil under eCO_2_ (Figure 4A–C).

### 3.6. Effects of AMF and CO_2_ on Soil Enzymes Activity

In general, eCO_2_ significantly increased the activity of soil invertase and urease (Figure 4D,E), but not the soil neutral phosphatase activity (Figure 4F) in both non-AMF and AMF plants. AMF colonization significantly increased the activity of invertase, urease, and neutral phosphatase (Figure 4D–F). Meanwhile, the CO_2_ × AMF interaction only resulted in significant positive changes in the invertase activity (Figure 4D), but not in both the urease (Figure 4E) and neutral phosphatase activity (Figure 4F). In addition, the activity of invertase, urease, and neutral phosphatase was significantly increased by 15–56%, 19–50%, and 15–39% in AMF than in non-AMF soil under ACO_2_, DeCO_2_, and (D + N)eCO_2_, but not under NeCO_2_ (Figure 4D–F).

### 3.7. Correlations

Total plant biomass production was significantly negatively correlated to soil AN (y = −0.10x + 12.08, Figure 5A) and soil AK (y = −0.06x + 17.42, Figure 5C) in non-AMF plants, but not to AMF plants (Figure 5A,C). In contrast, no relationships between total plant biomass production and soil AP were observed in both AMF and non-AMF plants (Figure 5B).

In addition, total plant biomass production was significantly positively correlated to the invertase activity in both AMF (y = 0.12x + 3.49) and non-AMF plants (y = 0.31x + 0.75, Figure 5D), to the urease activity in AMF plants (y = 1.30x + 2.44), but not to non-AMF plants (Figure 5E), and to the phosphatase activity in non-AMF plants (y = 16.81x + 2.01), but not to AMF plants (Figure 5F).

## 4. Discussion

### 4.1. Effects of AMF Symbiosis on Plant Biomass and C Accumulation Depend on eCO_2_ at Daytime or Nighttime

Studies on the understanding of mycorrhizal-CO_2_ responses are mostly focused on the differences between ACO_2_ and 550–1000 ppm eCO_2_ at daytime [17,19,55]; no information about mycorrhizal plants responses to eCO_2_ at nighttime has been reported. As a substrate for plant photosynthesis, eCO_2_ during daytime facilitates CO_2_ assimilation processes by increasing intercellular CO_2_ and leaf carboxylation efficiency of ribulose-1,5-bisphosphate carboxylase/oxygenase (RubisCO) while reducing photorespiration [6,56,57], leading to an accumulation of non-structural carbohydrates and a stimulation of biomass production (Figure 3A–F). Meanwhile, a part of the photosynthetically-fixed C is consumed by leaves, shoots, and roots through dark respiration [58]. As a product of plant respiration, an increased intercellular CO_2_ and reduced stomata conductance under nighttime eCO_2_ [59] would lead to a decrease of plant dark respiration [48]. As a result, more biomass production was found in wheat grown under 410/610 ppm NeCO_2_ than under 410/460 ppm ACO_2_ (Figure 3A–F), in line with earlier findings about Alfalfa [60], *Phaseolus vulgaris* [46], soybean [47], and *Xanthium strumarium* [44,47]. The decreased respiration was a physiological mechanism behind the increase in biomass by C conservation under NeCO_2_ [41,48,61].

Intriguingly, the most significant findings were that the pattern of plant response to eCO_2_ during daytime or nighttime was influenced by AMF inoculation. Plant growth depressions occur when increased nutrient benefits are outweighed by its C cost, whereas positive growth responses occur where benefits outweigh the cost [62]. The positive growth responses to AMF inoculation were observed under 550/460 ppm DeCO_2_ and 550/610 ppm (D + N)eCO_2_, whereas a negative effect under 410/610 ppm NeCO_2_ on wheat growth and C accumulation was found in the present study (Figure 3A–F). With low daytime CO_2_ and high nighttime CO_2_, photosynthetic responses did not predominately control plant growth [43]. AMF colonization would drain more photo-assimilates from the host plant for extraradical hyphal growth [25] and increase mycorrhizal respiration under eCO_2_ than under ACO_2_ [37,63], leading to reduced pools of nonstructural carbohydrates in the host and growth depression under NeCO_2_ (Figure 3A–F). With high CO_2_ concentration at daytime, plant C utilization by AMF might be compensated by higher photosynthesis in host plants [64]. AMF would enhance more photo-assimilate production than that they could drain from the host plant due to an improved nutrient uptake, leading to a positive effect under DeCO_2_ and (D + N)eCO_2_ (Figure 3A–F), which is in line with that reported by Zhu et al. [65], who concluded that AMF (*Rhizophagus irregularis*)-colonized wheat achieved greater growth and higher C accumulation than non-AMF wheat at 700 ppm (D + N)eCO_2_ [65]. Thus, resource limitation is a key factor in the cost–benefit analysis of AMF effects on plant growth under NeCO_2_.

### 4.2. Nitrogen Demands are Increased under eCO_2_, but AMF Symbiosis Lessens N Limitation under eCO_2_ at Daytime, Not at Nighttime

Our results showed that eCO_2_ at daytime and/or nighttime caused a decrease in N concentrations of the shoot and root both in non-AMF and AMF wheat (Table 1), which was similar to earlier reports that tissue N concentrations were often decreased in wheat cultivars under 550–800 ppm eCO_2_ [25,32,65,66,67]. The following mechanisms could explain the decreased N in plants under eCO_2_ both at daytime and nighttime: (1) a “dilution effect” due to higher plant biomass production [9,68]; (2) reduced transpiration rates under eCO_2_ both at daytime and nighttime could decrease the transpiration-driven mass flow of nutrients, and hence, induced limitations in leaf nutrient transport led to decreased N uptake [66,69,70]; and (3) the reduction of the RubisCO protein that constitutes about half of the protein in leaves under eCO_2_ both at daytime and nighttime [48,71]. Beyond that, we speculated that the inhibition of leaf N assimilation under eCO_2_ was associated with the reduction in respiration. The reduced respiration under high CO_2_ has led to a reduced supply of energy-rich compounds, including ATP and NADH in the cytoplasm, and thereby decreases the amount of reductant available for NO_3_^−^ reduction [72,73].

The N concentrations of shoots and roots and the P concentration of roots were higher in AMF plants than in non-AMF counterparts grown under DeCO_2_, whereas they were not affected by AM fungal colonization under NeCO_2_ (Table 1). NeCO_2_ hence resulted in a decrease in beneficial effects of AMF on plants. An enhanced C fixation under DeCO_2_ would require more supply of N, P and K, leading to a strong decline in soil AN, AP, and AK associated with DeCO_2_ (Figure 4A–C). Thus, *F. mosseae* colonization promoted plant uptake of N, P, and K, and alleviated plant nutrient demands and soil N, P, and K limitations under eCO_2_ at the daytime. In contrast, eCO_2_ at nighttime decreased the dark respiration, resulting in reduced energy and nutrient demands [44]. In general, plants in nutrient-rich soils under NeCO_2_ (Figure 4B,C) tend to be less frequently affected by AM fungi [74].

Although plant N concentrations were decreased due to CO_2_ elevation, N accumulation in wheat tissues was enhanced because of an increased biomass in both non-AMF and AMF plants (Figure 3G–I). Similar to our findings, compared to non-AMF plants, leaf N% or plant total N accumulation under 700–1000 ppm eCO_2_ was increased by 30–41% in AMF colonized wheat [65], alfalfa [36], and *Taraxacum officinale* [31]. The significantly higher N in AMF plants might be due to a higher C accumulation from photosynthesis and hence a greater N demand under daytime eCO_2_. In contrast, we found a negative AM effect on plant N uptake under nighttime eCO_2_ (Figure 3G–I). One possible explanation could be that eCO_2_ inhibited the assimilation of NO_3_^-^ [72], which is the dominant inorganic N form in dryland soils. The NO_3_^-^ assimilation of *Arabidopsis* and wheat was slowed down under 720 ppm nighttime eCO_2_ [72]. Thus, both the N form and atmospheric CO_2_ concentration at nighttime are important factors in determining plant performance.

### 4.3. P and K Demands were Increased under eCO_2_ at Daytime, Not at Nighttime

Shoot and root P or K concentrations were decreased under eCO_2_, regardless of AMF status (Table 1). These results agreed with how tissue P and K concentrations in various legumes and non-legumes were often lower under 550–800 ppm eCO_2_ than under ACO_2_ because of an increase in dry matter and carbohydrate accumulation [4,25,75,76]. Such lower P or K concentrations in plant tissues could be alleviated by AMF colonization in some, but not in all cases. For example, leaf and root P concentrations in non-mycorrhizal *T. repens* declined by 31% and 115% under 700 ppm eCO_2_, whereas in *F. mosseae* seedlings, the decline in P concentration was as low as 17% and 0.5%, respectively [77]. *R. intraradices* stimulated the growth and P acquisition of sour orange, but not of sweet orange grown at high P (2 mM) supply under 700 ppm eCO_2_ [37]. The growth of *Medicago truncatula* and *Brachypodium distachyon* under 900 ppm eCO_2_ was increased by sufficient P supply, rather than by *R. irregularis* colonization [9]. eCO_2_ at 700 ppm resulted in a 20% or 22% decrease of K in grains of non-mycorrhizal or mycorrhizal durum wheat [33]. In general, plants have developed specific P acquisition strategies by root systems and/or mycorrhizal associations to take up limited P in the soil [78,79,80]. Plant P and K acquisition is obviously enhanced by extensive root development. The P and K concentration influences plant photosynthesis and growth rates, leading to multiple C–P and C–K interactions 550 ppm DeCO_2_ (Figure 3J–O) while 610 ppm NeCO_2_ did not increase shoot, root, or total plant P and K content (Figure 3J–O). The different response to DeCO_2_ and NeCO_2_ may be related to photosynthesis. As a general rule, AMF symbiosis delivered soil N, P, and K to plants in return for photosynthate, alleviating plant N, P, and K demands, resulting in increased growth responses to eCO_2_ via positive feedback [3]. Thus, AMF symbiosis might alter the imbalance of sink and source under eCO_2_ by regulating the demand for C and supply of N, P, and K from the soil to the host plant [35]. However, our results showed that AMF colonization had no significant effects on P accumulations in the shoot and root and K accumulation in the root (Table 1, Figure 3J–L,N). The mechanisms affecting P, K absorption and metabolism in crops under AMF symbiosis and varied daytime and nighttime eCO_2_ needs further research in the future.

### 4.4. AMF Colonization Increased Soil N, P and K availability, Especially under eCO_2_ at Daytime

Plant–soil interactive feedback to eCO_2_ likely determines the increase or decrease in soil N, P, and K pools, as well as microbial community composition and activities. In general, C flows into the soil through the plant root and/or mycorrhiza directly or indirectly. Under eCO_2_, more C was available for mycorrhizal growth and development, and AMF associations were thus stimulated by such extra C [81]. In turn, AMF accelerates soil organic matter decomposition under eCO_2_ to mine for N and P [19]. The increase of invertase and urease activity under 500–610 ppm eCO_2_ indicated higher decomposition of organic C-N-compounds and release of N and other nutrients, resulting in a general activation of microbes [82]. The relative increases in soil enzyme activities also might be attributed to a higher biomass production under eCO_2_ (R^2^ = 0.46–0.77, *p* < 0.001, Figure 5A,D–F), and hence a greater demand of N, P, and K [3]. In fact, the considerable decrease in available N, P, and K under DeCO_2_ in the present study supports this view. A meta-analysis showed that eCO_2_-induced nutrient limitation could increase soil enzyme activities [83]. However, a significantly higher level of soil AN, AP and AK was observed in AMF wheat seedlings than in non-AMF counterparts under eCO_2_ (Figure 4A–C), suggesting that the presence of AM fungi could confer better soil fertility under eCO_2_.

## 5. Conclusions

The present study under auto-simulated future daytime and/or nighttime eCO_2_ provides evidence that eCO_2_ can impact plant–soil feedback and/or plant–AMF symbiosis. Our results showed that the responses of crops to an averaged sole daytime/nighttime eCO_2_ might not provide the expected effects of rising CO_2_ concentration that they could have on crop growth. More soil N, P, and K nutrients were required under DeCO_2_ to match increased C assimilation, leading to lower soil N, P, and K availability. Although *F. mosseae* colonization alleviated soil nutrient constraints in response to eCO_2_, its role on plant growth depended on eCO_2_ at daytime and/or nighttime, which could induce an imbalance in the source–sink relationship associated with reduced plant and soil N, P, and K. AMF symbiosis could improve plant C accumulation, N, P, and K uptake particularly under DeCO_2_, while NeCO_2_ decreased AMF’s beneficial effects on plants. Such information is essential for understanding the mechanisms influencing C, N, P, and K dynamics in future climate-change scenarios. As a result, integrating AMF’s benefits to plants under a factual field DeCO_2_ and NeCO_2_ will be critical when dealing with the long-term consequence of future CO_2_ rising on global cropping systems.

## Figures and Tables

**Figure 1 jof-07-00458-f001:**
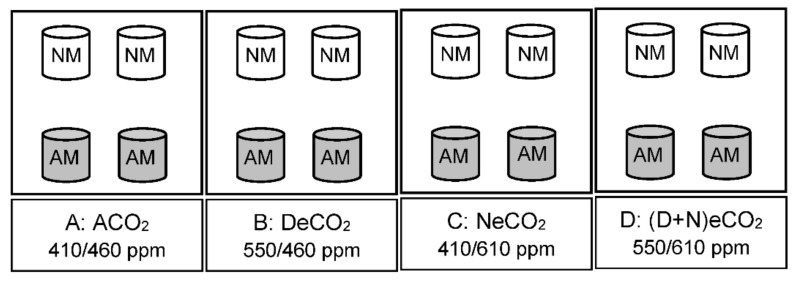
A schematic diagram showing the experiment designs. NM: non-AMF; AM: inoculated AMF. (**A**): ACO_2_, ambient CO_2_; (**B**): DeCO_2_, elevated CO_2_ concentrations at daytime; (**C**). NeCO_2_, elevated CO_2_ concentrations at nighttime; and (**D**): (D + N)eCO_2_, elevated CO_2_ concentrations at both daytime and nighttime.

**Figure 2 jof-07-00458-f002:**
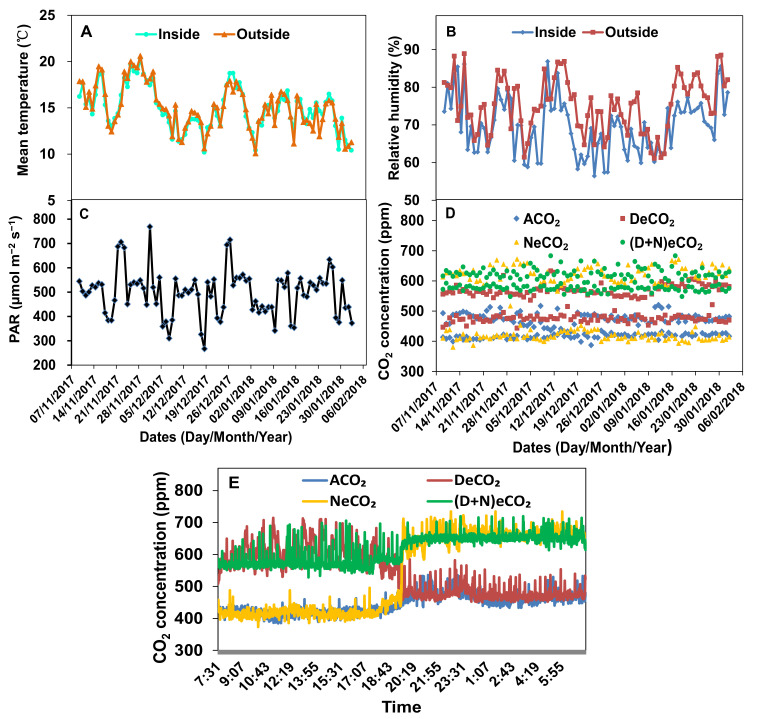
Mean temperature (**A**), relative humidity (**B**), photosynthetic active radiation (PAR, (**C**)) and CO_2_ concentration (**D**,**E**) over the experimental period in the growth chambers. ACO_2_: ambient CO_2_, DeCO_2_: elevated CO_2_ concentrations at daytime, NeCO_2_: elevated CO_2_ concentrations at nighttime, (D + N)eCO_2_: elevated CO_2_ concentrations at both daytime and nighttime.

**Figure 3 jof-07-00458-f003:**
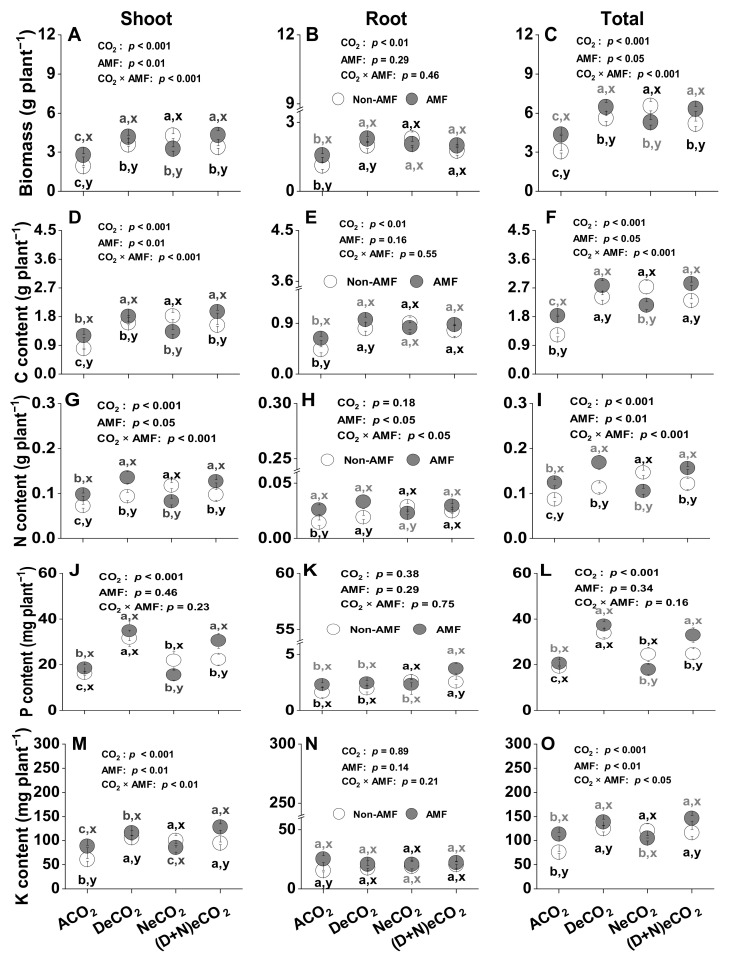
Effects of CO_2_ and mycorrhiza on (**A**) shoot, (**B**) root, and (**C**) total plant biomass production; (**D**) shoot C, (**E**) root C, and (**F**) total plant C accumulation; (**G**) shoot N, (**H**) root N, and (**I**) total plant N accumulation; (**J**) shoot P, (**K**) root P, (**L**) total plant P accumulation; and (**M**) shoot K, (**N**) root K, (**O**) total plant K accumulation in 12-week-old wheat grown under different daytime and/or nighttime CO_2_ concentrations inside environmentally controlled glass growth chambers. Values are the means ± standard error (SE), *n* = 3. Different letters above the bars indicate significant differences (*p* < 0.05), as revealed by Tukey’s test. Statistical comparisons (two-way ANOVA) between eCO_2_ or AMF treatments, as well as their eCO_2_ × AMF interaction are presented for each variable. Abbreviations: ACO_2_, ambient CO_2_ (410 ppm daytime + 460 ppm nighttime); DeCO_2_, elevated CO_2_ at daytime (550 ppm daytime + 460 ppm nighttime); NeCO_2_, elevated CO_2_ at nighttime (410 ppm daytime + 610 ppm nighttime); (D + N)eCO_2_, elevated CO_2_ at both daytime and nighttime (550 ppm daytime + 610 ppm nighttime). Daytime: 07:00 a.m.–19:00 p.m. and nighttime: 19:00 p.m.–07:00 a.m.

**Figure 4 jof-07-00458-f004:**
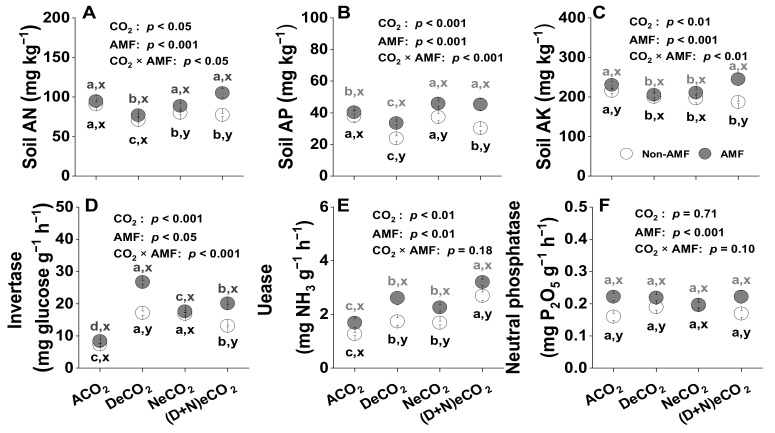
Effects of mycorrhiza and CO_2_ on (**A**) soil available nitrogen (AN), (**B**) soil available phosphorus (AP), (**C**) soil available potassium (AK), (**D**) invertase activity, (**E**) urease activity, and (**F**) neutral phosphatase activity in the soil of 12-week-old wheat grown under different daytime and/or nighttime CO_2_ concentrations inside environmentally controlled glass growth chambers. Values are the means ± SE, *n* = 3. Different letters above the bars indicate significant differences (*p* < 0.05), as revealed by Tukey’s test. Statistical comparisons (two-way ANOVA) between AMF and CO_2_ treatments, as well as their interaction (eCO_2_ × AMF) are presented for each variable. Abbreviations are the same as in Figure 2.

**Figure 5 jof-07-00458-f005:**
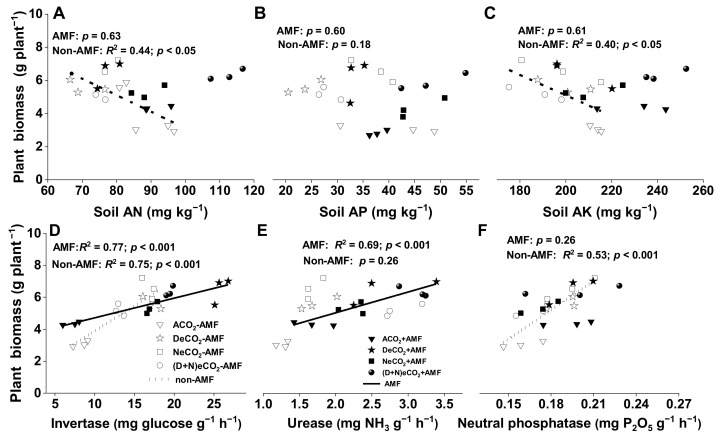
Relationships between plant biomass production and (**A**) soil available nitrogen (AN), (**B**) soil available phosphorus (AP), (**C**) soil available potassium (AK), (**D**) soil invertase, (**E**) urease or (**F**) neutral phosphatase in the soil of 12-week-old wheat grown under different daytime and/or nighttime CO_2_ concentrations inside environmentally controlled glass growth chambers. Data are means ± SE, *n* = 12. Regressions are shown for non-AMF (dotted lines) and for AMF (solid lines) treatments. Abbreviations are the same as in Figure 2.

**Table 1 jof-07-00458-t001:** Carbon, nitrogen, phosphorus and potassium concentrations in shoots and roots of non-AMF and AMF-inoculated (AMF) wheat plants grown under ambient CO_2_ (ACO_2_) and under-elevated CO_2_ concentrations at daytime (DeCO_2_), night (NeCO_2_), and both daytime and nighttime (D + N)eCO_2_.

Treatment		Carbon (mg g^−1^)		Nitrogen (mg g^−1^)		Phosphorus (mg g^−1^)		Potassium (mg g^−1^)	
Inoculation	CO_2_	Shoot	Root	Shoot	Root	Shoot	Root	Shoot	Root
Non-AMF	ACO_2_	413 ± 8 b,x	385 ± 9 b,x	37.4 ± 1.9 a,x	12.4 ± 1.1 a,y	8.39 ± 0.20 a,x	1.51 ± 0.07 a,x	31.49 ± 0.66 a,x	13.05 ± 0.96 a,y
	DeCO_2_	445 ± 10 a,x	403 ± 12 b,x	26.0 ± 2.3 b,y	9.5 ± 0.5 b,y	8.80 ± 1.05 a,x	0.96 ± 0.14 b,y	29.66 ± 0.81b,x	8.72 ± 1.21 b,x
	NeCO_2_	430 ± 9 a,x	393 ± 14 b,x	27.6 ± 0.8 b,x	12.6 ± 0.3 a,x	5.15 ± 0.84 b,x	1.17 ± 0.11 b,x	23.60 ± 1.82 d,y	8.56 ± 1.17 b,x
	(D + N)eCO_2_	449 ± 13 a,x	432 ± 16 a, x	28.3 ± 0.8 b,x	14.0 ± 1.0 a,x	6.46 ± 1.34 b,x	1.38 ± 0.06 a,y	27.76 ± 0.70c,y	11.43 ± 1.04 a,x
AMF	ACO_2_	432 ± 7 b,x	405 ± 15 b,x	35.2 ± 0.7 a,x	16.8 ± 0.9 a,x	6.67 ± 0.55 b,y	1.47 ± 0.13 b,x	31.84 ± 0.61 a,x	15.89 ±0.63 a,x
	DeCO_2_	434 ± 8 b,x	419 ± 7 b,x	32.5 ± 1.2 b,x	14.3 ± 0.9 b,x	8.42 ± 0.63 a,x	1.17 ± 0.16 c,x	28.08 ± 1.06 b,x	9.03 ± 1.15 b,x
	NeCO_2_	410 ± 11 c,x	400 ± 18 b,x	25.3 ± 1.0 c,x	11.5 ± 1.5 b,x	4.73 ± 0.47 c,x	1.09 ± 0.15 c,x	26.20 ± 1.64 b,x	9.45 ± 0.98 b,x
	(D + N)eCO_2_	454 ± 12 a,x	438 ± 10 a,x	29.5 ± 1.4 b,x	14.9 ± 0.8 b,x	7.03 ± 0.49 b,x	1.84 ± 0.19 a,x	29.67 ± 0.82 b,x	9.79 ± 0.72 b,x
ANOVA									
CO_2_		*	*	***	*	**	*	**	*
AMF		ns	ns	ns	*	ns	ns	ns	ns
eCO_2_×AMF		ns	ns	*	*	ns	ns	ns	*

Data (means ± SE, *n* = 3) followed by different letters indicate significant differences between CO_2_ treatments for the same AMF inoculation (a, b, c, d) and between AMF inoculations for the same CO_2_ treatment (x, y) at *p* < 0.05. ANOVA: ns not significant; *, ** and *** indicate significant differences at *p* ≤ 0.05, *p* ≤ 0.01 and *p* ≤ 0.001, respectively.

## Data Availability

Not applicable.

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
