# Peer review of "Funneliformis mosseae Improves Growth and Nutrient Accumulation in Wheat by Facilitating Soil Nutrient Uptake under Elevated CO2 at Daytime, Not Nighttime"

_jof, 2021, doi:10.3390/jof7060458_

Round 1
Reviewer 1 Report
The manuscript by Shi et al aims at disentangling how elevated CO2 (eCO2) and mycorrhization affect wheat (Triticum aestivum) performance and soil fertility. To this end, wheat seedlings were grown under four different CO2 environments for 12-weeks, that were (1) ACO2 (410/460 ppm, daytime), (2) sole daytime eCO2 (DeCO2, 550/460 ppm), (3) sole nighttime eCO2 (NeCO2, 410/610 ppm), and (4) dual or continuous daytime/nighttime eCO2 ((D+N)eCO2, 550/610 ppm), and with or without AMF (Funneliformis mosseae) colonization. The authors measured several parameters in plants such as the level of mycorrhizal colonization, the biomass and the N, P, K contents. They measured also available N, P and K in soil, as well as invertase, urease and phosphatase activities. The results obtained showed that AMF colonization alleviated soil nutrient constraints on plant responses to DeCO2, while NeCO2 decreased AMF’s beneficial effects on plants.
Overall, the manuscript is well written and I have only minor comments that are listed below:
My main comment is that despite the work has been well conducted I found that the number of replicates was very low (n=3). The authors wrote that they had six pots for each CO2 treatment but they pooled 2 pots to get one composite sample. Could the authors explain better why they did that ?
In the Material and Methods section, the authors cite at least 3 papers (51, 52, 53) that are impossible to find. Would it be possible to give other references (more accessible) for the methods that have been used?
In addition, part 2.7 is repetitive, please rewrite it by putting the common methodology once and then the specific ones.
Figure 2: the significance of the symbols is missing. I supposed that empty circles were for non AMF and the grey ones for AMF. Could you add this information, please?
Author Response
My main comment is that despite the work has been well conducted I found that the number of replicates was very low (n=3). The authors wrote that they had six pots for each CO2 treatment but they pooled 2 pots to get one composite sample. Could the authors explain better why they did that?
Response: Thanks for your comments. The number of replicates was three since the construction of the CO2 auto-controlling system and environmentally controlled chambers were very high costs. As a result, we constricted a total of 12 growth chambers for four CO2 concentration treatments (each treatment with three chambers or replicates). To minimize differences in plant performance, each growth chamber or replicate had two mycorrhizal pots and two non-mycorrhizal treatment pots, respectively. Thus the three growth chambers or replicates had a total of six replicated pots for each CO2 concentration treatment. Plants or soils from these two pots in each chamber or replicate were collected and combined as one composite sample, while the three composite samples (n = 3) were from six pots for each CO2 treatment.
In the Material and Methods section, the authors cite at least 3 papers (51, 52, 53) that are impossible to find. Would it be possible to give other references (more accessible) for the methods that have been used?
Response: Thanks for your comments. This study adopted the root AMF colonization determination method of Brundrett M, Bougher N, Dell B, Grove T, Malajczuk N. 1996. Working with Mycorrhizas in Forestry and Agriculture (Australian Centre for International Agricultural Research Monograph 32, 173-212), Canberra, Australia. As today of 04 June 2021, this paper of Brundrett et al. (1996) has 1371 citations, which is a worldwide used experimentation manual for mycorrhizal research. Dr. Mark Brundrett, a colleague of the corresponding author in this study at the University of Western Australia, kindly to allow every interested personnel to refer to “http://mycorrhizas.info/info.html”, which contains almost all latest information on running mycorrhizal experiments.
The route determination of C, N, P and K concentrations in plant and soil were according to Yang et al. (2008), which is a national textbook widely used in China.
The reference about the assays of soil enzyme activities of “Guan, S. Y.; Zhang, D.; Zhang, Z., Soil enzyme and its research methods. Agricultural Press, Beijing: 1986,274–340” has been changed as follows (please also see the Ref 53-54 in the revised version”.
Ren, C.; Kang, D.; Wu, J. p.; Zhao, F.; Yang, G.; Han, X.; Feng, Y.; Ren, G., Temporal variation in soil enzyme activities after afforestation in the Loess Plateau, China. Geoderma 2016, 282, 103-111.
Wu, Q. S.; Li, Y.; Zou, Y. N.; He, X. H., Arbuscular mycorrhiza mediates glomalin-related soil protein production and soil enzyme activities in the rhizosphere of trifoliate orange grown under different P levels. Mycorrhiza 2015, 25, (2), 121-30.
In addition, part 2.7 is repetitive, please rewrite it by putting the common methodology once and then the specific ones.
Response: Thanks for your comments. The writings of part 2.7 of
“Soil invertase activity (mg of glucose released per g dry soil per 24 h) was determined using 8 % (w/v) glucose solution as substrate. 5 g fresh soil was incubated for 24 h at 37 °C with 1 ml toluene, 15 ml 8% (w/v) sucrose and 5 ml phosphate buffer (pH 5.5). After incubation, the mixture was immediately filtered, and then 1 ml filtrate with 3 ml 3,5-dinitrosalicylic acid were taken and incubated in boiling water for 5 min. Subsequently, reaction solution was diluted to 50 ml with distilled water and measured by spectrophotometry at 508 nm .
Soil urease activity (mg of NH4+–N released per g soil per 24 h) was determined using 10 % (w/v) urea solution as substrates. 5 g fresh soil was incubated for 24 h at 37 °C with 1 ml toluene, 10 mL 10% urea solution (w/v) and 20 ml citrate buffer (pH 6.7). After incubation, the mixture was immediately filtered, and then 2 ml filtrate, 4 ml sodium phenol solution and 3 ml 0.9 % (v/v) sodium hypochlorite solution was taken into 50 ml volumetric flask and mixed. 20 min later, reaction solution was dilute to 50 ml with distilled water and measured by spectrophotometry at 578 nm.
Neutral phosphatase activity (mg of phenol per g soil per 24 h) was determined by incubating 5.0 g fresh soil, 1 ml toluene and 5 ml disodium phenyl phosphate solution and 5 ml citrate buffer (pH 7.0) for 24 h at 37 °C. After incubation, the mixtures were immediately filtered, and then 1 ml filtrate, 5 ml borate buffer (pH 9.0), 3 ml 2.5 % potassium ferrocyanide (w/v) and 3 ml 0.5 % 4-aminoantipyrine (w/v) were taken into 50 ml volumetric flask and mixed. Subsequently, reaction solution was dilute to 50 ml with distilled water and measured by spectrophotometry at 570 nm .” have been changed to
“Soil invertase activity (mg glucose g-1 h-1) was determined firstly by incubating five grams of fresh soil with 1.0 ml toluene, 15 ml 8% (w/v) sucrose and 5 ml phosphate buffer (pH 5.5) for 24 h at 37°C. After incubation, 1 ml filtrate with 3 ml 3,5-dinitrosalicylic acid were then incubated in boiling water for 5 min. Subsequently, the reaction solution was diluted to 50 ml with distilled water and spectrophotometrically measured at 508 nm [53].
Soil urease activity (mg NH4+–N g-1 h-1) was determined firstly by incubating five grams of fresh soil with 1 ml toluene, 10 ml 10% urea (w/v) and 20 ml citrate buffer (pH 6.7) for 24 h at 37°C. After incubation, 2 ml filtrates were mixed with 4 ml sodium phenol and 3 ml 0.9% (v/v) sodium hypochlorite solution in in a 50 ml volumetric flask. After 20 min, the reaction solution was then diluted to 50 ml with distilled water and spectrophotometrically measured at 578 nm [53].
Neutral phosphatase activity (mg phenol g-1 h-1) was determined firstly by incubating five grams of fresh soil with 1 ml toluene and 5 ml disodium phenyl phosphate and 5 ml citrate buffer (pH 7.0) for 24 h at 37°C. After incubation, 1 ml filtrate with 5 ml borate buffer (pH 9.0), 3 ml 2.5% potassium ferrocyanide (w/v) and 3 ml 0.5% 4-aminoantipyrine (w/v) were then thoroughly mixed in a 50 ml volumetric flask. Subsequently, the reaction solution was dilute to 50 ml with distilled water and spectrophotometrically measured at 570 nm [54].”
Figure 2: the significance of the symbols is missing. I supposed that empty circles were for non AMF and the grey ones for AMF. Could you add this information, please?
Response: Thanks for your comments. To keep the figure not so crowded, the legends were only displayed in the Figure 2B. We have also now added the legends in the Figure 2E, 2H, 2K and 2N.

Reviewer 2 Report
The manuscript entitled “Funneliformis mosseae improves growth and nutrient accumulation in wheat by facilitating soil nutrient uptake under elevated CO2 at daytime, not nighttime” by Shi et al. found that arbuscular mycorrhizal fungi (AMF) and elevated carbon dioxide (CO2) interactively affected wheat growth and nutrient uptake. The authors found that both elevated CO2 and application of AMF positive regulated plant growth and nutrient uptake. Detail investigations suggested that elevated CO2 at daytime is more important that those at nighttime, suggested the important of photosynthesis might be the main mechanism. Additionally, this manuscript also suggested some possible management consideration for the long-term consequence CO2 rising on global cropping systems in the future. However, there are some format errors in the text and figures. Therefore, some major and minor modifications are still needed to improve the quality of the manuscript as following.
Major comments
I suggest to make a figure to show the experiment design (Eight treatments). In this case, the reader will be easy to understand the purpose and results of this paper.
The language still not good in some sentences. It should be modified by some authority institute or English native speakers.
Minor comments
- Line 17. I suggest to give a brief background of this research, in an declarative sentence not and interrogative sentence with a question. I also suggest to replace “different atmosphere CO2 (ACO2) concentrations” by “elevated CO2”. Because the manuscript did not touch on the decreased atmosphere CO2.
- Line 27. Please replace “Compared to” by “Compared with”.
- Line 39. “The 280 ppm atmospheric CO2 concentration (ACO2) during the industrial revolution has been increased to 414.49 ppm in December 2020, and could be reached to ~550 ppm in the next 50 years based on an annual 2.40 ppm CO2 growth rate from 2010 to 2020.” This sentence is not good. Please rephase this sentence.
- Line 44. Please replace “eCO2” by “Elevated CO2”, if this word was used in the begin of the sentence.
- Line 46. Please provide Latin name for “wheat”, when you mention in the fist time. And did not mention the Latin name in the later text. For the word “more”, if you use this, another word “than” or “comparing” should appear. In this case, it will be “English”.
- Line 53. Please remove “have” here.
- Line 65. Please replace “AM” by “AMF”.
- Line 95. Please provide Latin name for “mulberry”.
- Line 100. Please replace “20–45 %” by “20–45%”.
- Line 117. Please replace “(Triticum aestivum Yunmai)” by “(T. aestivum cv. Yunmai)”.
- Line 140. Please provide Latin name for those species, “sorghum, soybean” if they were mentioned for the first time.
- Line 143. Please replace “The humidity….” by “The air humidity….” .
- Line 148. Please confirm the unit of temperature and there is no space between number and “%”.
- Figure 1. I suggest to use “The day of year (DOY)” not date in the X-axis of A-D. Please also provide full names for “ACO2, DeCO2, NeCO2 and (D+N)eCO2” in the figure legend.
- Line 191. Please replace “(Triticum aestivum Yunmai)” by “(T. aestivum cv. Yunmai)”. Please confirm the concentration of hydrogen peroxide (H2O2), 10%? It seems too high.
- Line 215. Please confirm the word “50 %” and “AM”. “50%” and “AMF”?
- Line 244. I suggest to replace “5 g….” by “Five g…..”.
- Line 274, and Line 281. Here you used “greater” and “higher”, by comping with what? Please rephase those sentences.
- Figure 2. Please use same font-size in one figure. It seems the font-size of M, N, P are smaller than others.
- Line 350. Please replace “was” by “were”.
- Figure 4. Please use same font-size in one figure. It seems the font-size of D-F are smaller than others.
- Line 390-391. The “……AN (y=-0.10x + 12.08, Figure 4A) and soil AK (y =-0.06x + 17.42, Figure 4C)…… ” should be “……AN (y = -0.10x + 12.08, Figure 4A) and soil AK (y = -0.06x + 17.42, Figure 4C)……”.
- Line 532. Please replace “….on response to….” by “….in response to….”.
- Line 535. Please replace “AM symbiosis….” by “AMF symbiosis….”.
Author Response
Major comments
I suggest to make a figure to show the experiment design (Eight treatments). In this case, the reader will be easy to understand the purpose and results of this paper.
Response: Thanks for your suggestion. We have now added the new Figure 1 to show the experiment design in the revised version.
The language still not good in some sentences. It should be modified by some authority institute or English native speakers.
Response: Thanks for your comment. Professor Xinhua He, who has been working at the University of California at Davis and the University of Western Australia for > 25 years, has again checked the science and language throughout the whole revised version.
Minor comments
- Line 17. I suggest to give a brief background of this research, in an declarative sentence not and interrogative sentence with a question. I also suggest to replace “different atmosphere CO2(ACO2) concentrations” by “elevated CO2”. Because the manuscript did not touch on the decreased atmosphere CO2.
Response: Thanks for your comments. The sentence of “How different atmosphere CO2 (ACO2) concentrations and arbuscular mycorrhizal fungi (AMF) could concurrently affect plant growth” has been changed to “The concurrent effect of elevated CO2 (eCO2) concentrations and arbuscular mycorrhizal fungi (AMF) on plant growth”.
- Line 27. Please replace “Compared to” by “Compared with”.
Response: Done as suggested, thanks.
- Line 39. “The 280 ppm atmospheric CO2 concentration (ACO2) during the industrial revolution has been increased to 414.49 ppm in December 2020, and could be reached to ~550 ppm in the next 50 years based on an annual 2.40 ppm CO2 growth rate from 2010 to 2020.” This sentence is not good. Please rephase this sentence.
Response: Thanks for your comments. The sentence of “The 280 ppm atmospheric CO2 concentration (ACO2) during the industrial revolution has been increased to 414.49 ppm in December 2020, and could be reached to ~550 ppm in the next 50 years based on an annual 2.40 ppm CO2 growth rate from 2010 to 2020.” has been changed to “The ACO2 concentration has been increased from 280 ppm during the industrial revolution to 419 ppm in April 2021 (https://www.co2.earth), and could be reached to ~550 ppm in the next 50 years”.
- Line 44. Please replace “eCO2” by “Elevated CO2”, if this word was used in the begin of the sentence.
Response: Done as suggested, thanks.
- Line 46. Please provide Latin name for “wheat”, when you mention in the fist time. And did not mention the Latin name in the later text. For the word “more”, if you use this, another word “than” or “comparing” should appear. In this case, it will be “English”.
Response: Thanks for your comments. The Latin name of “wheat” has been added, and the “more” has been changed to “a large”.
- Line 53. Please remove “have” here.
Response: Done as suggested, thanks.
- Line 65. Please replace “AM” by “AMF”.
Response: Done as suggested, thanks.
- Line 95. Please provide Latin name for “mulberry”.
Response: Done as suggested, thanks.
- Line 100. Please replace “20–45 %” by “20–45%”.
Response: All of these formats have been corrected as suggested, thanks..
- Line 117. Please replace “(Triticum aestivumYunmai)” by “( aestivum cv. Yunmai)”.
Response: Done as suggested, thanks.
- Line 140. Please provide Latin name for those species, “sorghum, soybean” if they were mentioned for the first time.
Response: Done as suggested, thanks.
- Line 143. Please replace “The humidity….” by “The air humidity….” .
Response: Done as suggested, thanks.
- Line 148. Please confirm the unit of temperature and there is no space between number and “%”.
Response: All of these formats have been corrected as suggested, thanks..
- Figure 1. I suggest to use “The day of year (DOY)” not date in the X-axis of A-D. Please also provide full names for “ACO2, DeCO2, NeCO2 and (D+N)eCO2” in the figure legend.
Response: Thanks for your comments. The X-axis of Figure 1A-D have been changed to “Dates (Day/Month/Year)”, and full names of “ACO2, DeCO2, NeCO2 and (D+N)eCO2” have been added in the Line 167-169 in the revised version.
- Line 191. Please replace “(Triticum aestivum Yunmai)” by “( aestivum cv. Yunmai)”. Please confirm the concentration of hydrogen peroxide (H2O2), 10%? It seems too high.
Response: Thanks for your comments. The writing of “Seeds of winter wheat (T aestivum cv. Yunmai) were sterilized with 10 % H2O2 for 20 min” has been changed to “Seeds of winter wheat (T. aestivum cv. Yunmai) were surface sterilized with 10% H2O2 for 20 min”.
- Line 215. Please confirm the word “50 %” and “AM”. “50%” and “AMF”?
Response: Thanks for your comments. It should be “50%” and “AMF”.
- Line 244. I suggest to replace “5 g….” by “Five g…..”.
Response: Thanks for your comments. The “5 g” in Line 244, 251 and 258 have been changed to “Five grams of”.
- Line 274, and Line 281. Here you used “greater” and “higher”, by comping with what? Please rephase those sentences.
Response: Thanks for your comments. The writing in Line 274 “Significantly greater percentage of root AMF colonization among CO2 treatments (P < 0.05) in AMF plants was ranked as” has been changed to “Significantly greater percentage of root AMF colonization among CO2 treatments (P < 0.05) ranked in AMF plants as”.
The writing in Line 281 “significantly higher shoot C concentrations in AMF plants ranked as (D+N)eCO2 > DeCO2 ≈ ACO2 > NeCO2” has been changed to “significantly higher shoot C concentrations ranked in AMF plants as (D+N)eCO2 > DeCO2 ≈ ACO2 > NeCO2”.
- Figure 2. Please use same font-size in one figure. It seems the font-size of M, N, P are smaller than others.
Response: Done as suggested, thanks.
- Line 350. Please replace “was” by “were”.
Response: Done as suggested, thanks.
- Figure 4. Please use same font-size in one figure. It seems the font-size of D-F are smaller than others.
Response: Done as suggested, thanks.
- Line 390-391. The “……AN (y=-0.10x + 12.08, Figure 4A) and soil AK (y =-0.06x + 17.42, Figure 4C)…… ” should be “……AN (y = -0.10x + 12.08, Figure 4A) and soil AK (y = -0.06x + 17.42, Figure 4C)……”.
Response: Thanks for your comments. Done as suggested, thanks.
- Line 532. Please replace “….on response to….” by “….in response to….”.
Response: Done as suggested, thanks.
- Line 535. Please replace “AM symbiosis….” by “AMF symbiosis….”.
Response: Done as suggested, thanks.
